# Radiomics and Molecular Classification in Endometrial Cancer (The ROME Study): A Step Forward to a Simplified Precision Medicine

**DOI:** 10.3390/healthcare10122464

**Published:** 2022-12-07

**Authors:** Giorgio Bogani, Valentina Chiappa, Salvatore Lopez, Christian Salvatore, Matteo Interlenghi, Ottavia D’Oria, Andrea Giannini, Umberto Leone Roberti Maggiore, Giulia Chiarello, Simona Palladino, Ludovica Spano’ Bascio, Isabella Castiglioni, Francesco Raspagliesi

**Affiliations:** 1Fondazione IRCCS Istituto Nazionele dei Tumori di Milano, 20121 Milano, Italy; salvatore.lopez@istitutotumori.mi.it (S.L.); ulrm@me.com (U.L.R.M.); giulia.chiarello@istitutotumori.mi.it (G.C.); simona.palladino@istitutotumori.mi.it (S.P.); ludovica.spanobascio@istitutotumori.mi.it (L.S.B.); francesco.raspagliesi@istitutotumori.mi.it (F.R.); 2DeepTrace Technologies, Scuola Universitaria Degli Studi Superiori IUSS-Pavia, 27100 Pavia, Italy; salvatore@deeptracetech.com (C.S.); interlenghi@deeptracetech.com (M.I.); 3Department of Medical and Surgical Sciences and Translational Medicine, Sapienza University, 00185 Rome, Italy; ottavia.doria@uniroma1.it (O.D.); andrea.giannini@uniroma1.it (A.G.); 4Dipartimento di Fisica G. Occhialini, University of Milan-Bicocca, 20121 Milano, Italy; castiglioni.isabella@gmail.com

**Keywords:** endometrial cancer, radiomics, molecular, genomic, radiogenomics, ultrasound

## Abstract

Molecular/genomic profiling is the most accurate method to assess prognosis of endometrial cancer patients. Radiomic profiling allows for the extraction of mineable high-dimensional data from clinical radiological images, thus providing noteworthy information regarding tumor tissues. Interestingly, the adoption of radiomics shows important results for screening, diagnosis and prognosis, across various radiological systems and oncologic specialties. The central hypothesis of the prospective trial is that combining radiomic features with molecular features might allow for the identification of various classes of risks for endometrial cancer, e.g., predicting unfavorable molecular/genomic profiling. The rationale for the proposed research is that once validated, radiomics applied to ultrasonographic images would be an effective, innovative and inexpensive method for tailoring operative and postoperative treatment modalities in endometrial cancer. Patients with newly diagnosed endometrial cancer will have ultrasonographic evaluation and radiomic analysis of the ultrasonographic images. We will correlate radiomic features with molecular/genomic profiling to classify prognosis.

## 1. Introduction

Endometrial cancer is one of the most common gynecological malignancies in developed countries [1]. In 2022, approximately 66,000 new endometrial cancer cases and 12,500 cancer-related deaths are projected to occur in the United States (U.S.) [1]. Traditionally, endometrial cancer is considered an indolent disease [2]. The majority of patients are diagnosed in the early stages of the disease and with endometrioid endometrial cancer [2]. Traditionally, histology and disease extension are the main factors predicting survival [2].

Recent evidence suggests that the evaluation of molecular and genomic profiling provides an accurate method to assess the prognosis of endometrial cancer patients [3,4]. In 2013, The Cancer Genome Atlas (TCGA) Research Network identified at least four groups of endometrial cancer patients, characterized by different prognoses (POLE-mutated or ultra-mutated; microsatellite instability or hypermutated; copy number low; copy number high) [3,4,5]. 

The Proactive Molecular Risk Classifier for Endometrial Cancer (ProMisE) classification simplified the clustering of patients into four categories characterized by different mutational patterns [3]. According to the ProMiseE endometrial cancer patients might be classified in four subgroups: (i) POLE-mutated; (ii) mismatch-repair deficiency; (iii) p53 wild type; and (iv) p53 abnormal. 

The adoption of molecular and genomic profiling provides useful information for prognostication and for tailoring the need for adjuvant therapies [3,5]. However, Next Generation Sequencing (NGS) needed to perform molecular evaluation is costly and characterized by a not negligible turnaround time. To overcome issues related to the time needed for analysis and the cost-effectiveness of molecular/genetic profiling, other sustainable methods can be assessed as alternative clinical tools. Accumulating data, published in collaboration with our study group, underlines the fact that various ultrasonographic patterns (assessed with transvaginal ultrasound) might be useful in identifying low- and high-risk endometrial cancer according to their morphological characteristics [6,7]. 

Radiomics is a method that analyzes a large number of quantitative features from radiological images using data-characterization algorithms. The information extracted is a measurement of tissue heterogeneity reflecting the underlying pathophysiology that might be correlated with clinical data and with information on mutational status (i.e., radiogenomics) [8,9,10,11] explaining different prognostic phenotypes. The application of radiomics shows important results, across various radiological images, including Ultrasonography (US) and oncologic specialties [8,9]. Interestingly, transvaginal ultrasound, Computed Tomography (CT) scan and magnetic resonance imaging are used to assess local, loco-regional, and distant spread [12,13]. Interestingly, other authors applied radiomics to MRI with exciting results [13]. There is a great need to obtain earlier and more biologically informative data from endometrial tumors that could assist in defining prognosis and planning the optimal course of treatment for women. Here, we aim to compare data obtained from radiomic analysis to data from molecular and genomic profiling. This process will provide a new toll for the characterization of endometrial cancer. 

## 2. Materials and Methods

This protocol was approved and financed by the Italian Minister of Health and from the Fondazione IRCCS Istituto Nazionale dei Tumori di Milano (Milano, Italy). The project won the “Ricerca Finalizzata 2019” grant. The Fondazione IRCCS Istituto Nazionale dei Tumori di Milano financed this protocol under the clinical study protocol “Correlation between Radiomic and Molecular Classification in Endometrial Cancer (RACE)”. The Institutional Review Board (IRB) was obtained (IRB#140/20; approved on 30 June 2020). The main outcome measure is to test concordance between radiomic features, pathological variables and molecular/genomic profiling. All consecutive patients with newly diagnosed endometrial cancer were enrolled starting on 1 January 2021. Inclusion criteria will be: (i) preoperative endometrial cancer (any histology); (ii) preoperative ultrasonographic examination; (iii) surgical treatment; (iv) age ≥ 18 years; and (v) signed informed consent for research purposes. Exclusion criteria will be: (i) consent withdrawn; (ii) lack of available (tumor) tissue for molecular/genomic profiling.

### 2.1. Specific Aims and Experimental Design

Specific Aim 1: Evaluating the predictive value of radiomic features in predicting specific mutational patterns in endometrial cancer patients.

Specific Aim 2: Evaluating the scaled impact of radiomic signals.

Specific Aim 3: Assessing the intra-observer and inter-observer variability of radiomic signals obtained during clinical preoperative workup.

Experimental Design Aim 1: We will perform both radiomic analysis and molecular/genomic profiling in a cohort of 100 consecutive endometrial cancer patients (see sample size calculation located in “statistical analysis” section, reported below). All images will be selected and stored by an expert ultrasonographer. Ultrasonographic examination and data collection will be standardized, thus avoiding possible confounding factors. Images will be analyzed. Then, we will correlate radiomic findings with molecular/genomic profiling to confirm and assess how radiomic signals predict high-risk disease.

Experimental Design Aim 2: We will focus our analysis on cases with discordant results between radiomic analysis and molecular/genomic profiling. Retrospectively, we will investigate new signals that might be useful in improving concordance between these two methods.

Experimental Design Aim 3: The study will include a validation cohort of 40 patients. This cohort of patients will be used to test the intra-observer and interobserver agreement. Ultrasonographic evaluation and image selection of the tumors will be performed by three members of the team (including the expert ultrasonographer and two investigators who have no extensive background in ultrasonographic examination) to assess the reproducibility of our findings.

### 2.2. Methodology

The gynecologic oncology unit of the Fondazione IRCCS Istituto Nazionale dei Tumori di Milano is a referral center for the treatment of endometrial cancer. Approximately, 100 endometrial cancer patients receive treatment in our unit, per year. For this project, we plan to enroll all consecutive endometrial cancer patients having surgery at our unit. Preoperative workup will include the execution of transvaginal ultrasound for all patients. During this process, ultrasonographic images of the endometrial tumor will be acquired. The same trained ultrasonographer (VC) will evaluate all patients. Ultrasonographic evaluation and image selection are standardized according to IETA guidelines [7]. Ultrasound will be carried out with the same Samsung Hera w10 instrumentation. The analysis will be performed in the same room. Room temperature will be checked continuously to avoid possible confounding factors. We will test the intra-observer and inter-observer agreement of ultrasonographic analyses. Radiomic analysis will be carried out prospectively before surgery. The PI (GB) and Co-PI (VC) will carry out the analysis. The radiomic analysis will be performed using TRACE4© radiomic platform (http://www.deeptracetech.com/files/TechnicalSheet__TRACE4.pdf (URL accessed 12 December 2022)), including radiomic classifiers developed for ultrasonography and gynecologic cancers [14]. The radiomics methodology will be applied to collected ultrasonographic images of patients, according to the International Biomarker Standardization Initiative (IBSI) guidelines (https://arxiv.org/abs/1612.07003 URL accessed 12 December 2022)) [15]. Details of radiomics analysis is reported elsewhere [14,16]. All patients included will be submitted to surgical treatment. Surgical treatment will include laparoscopic hysterectomy, bilateral salpingo-oophorectomy, and sentinel node mapping. Other surgical procedures will be carried out on the basis of patients’ and disease characteristics, as reported elsewhere [15,16]. Then, after surgery, molecular/genomic profiling will be performed by a highly experienced member of the team (SL) using Illumina^®^ (Illumina sequencing technology). Results achieved from radiomic and molecular/genomic profiling will be compared to identify specific signals that are predictive of patients’ outcomes. The members of the team with an extensive background in biostatics and radiomics (GB, SL, VC, CS and IC) will perform all analyses, and correlate radiomic signals with molecular/genomic profiling. A radiomic classifier of risk of endometrial cancer will be implemented by the platform TRACE4.

### 
2.3. Statistical Analysis


The sample size is calculated according to Hajian-Tilaki [17] analysis for diagnostic studies: assuming a sensitivity of 98.5% in predicting high-risk disease and high-risk prevalence of 15%, a precision of estimate (i.e, the maximum marginal error) *d* = 5%, and a type I error alpha = 0.05, a sample size of 86 patients is needed to test the general hypothesis (i.e., to answer whether radiomics will accurately predict unfavorable molecular/genomic profiling). Assuming a dropout rate of 10%, a total of 100 patients will be enrolled in the study. Additionally, a further 40 patients will be used as a validation cohort.

## 3. Discussion

Accumulating evidence highlights the fact that traditional histological features are inaccurate in terms of predicting the risk of recurrence and survival outcomes of endometrial cancer patients. To date, molecular/genomic profiling executed after surgery is the most accurate method to assess patients’ risk and the need for further surgical and adjuvant treatments [16,17,18,19,20,21]. However, radiomics is an emerging tool applied to radiological images that is able to capture the heterogeneity of phenotypes leading to different prognoses. The adoption of preoperative radiomic analysis on ultrasound images of endometrium associated with molecular portrait will allow for the planning of personalized surgical and adjuvant treatments for endometrial cancer patients, according to precision medicine [22]. Moreover, thanks to such association, in vivo radiomics will replace the need for extensive ex vivo molecular/genomic profiling of endometrial cancer. Radiomic data will be harnessed through quantitative image analysis and leveraged via clinical decision support systems to improve decision making processes in patients at risk of endometrial cancer. 

In this project, we will enroll all consecutive patients with newly diagnosed endometrial cancer, regardless of histology types and stages at presentation. The radiomic analysis would be performed on primary (intrauterine) tumor. The first cohort of 100 consecutive patients will be submitted to preoperative ultrasonographic evaluation. Images will be stored. We will perform a radiomic analysis of ultrasonographic images with a radiomic software platform compliant to radiomic standards (IBSI). During the first 18 months, we plan to complete the analysis of radiomic features and the correlation between radiomics signals and molecular/genetic profiling, defining at least two classes of risks, e.g., favorable-unfavorable molecular/genomic profiling. In the second phase of the study, we will focus on re-analyzing data of patients with discordant results between radiomic and molecular/genetic profiling to identify new radiogenomic signals predictive of patients’ risk. Additionally, a validation cohort of patients will be used to assess intra- and inter-observer variability for the process of image acquisition and radiomic analysis.

Possible pitfalls of the project include the following: (i) radiomic features variability: the complexity of radiomic/radiogenomic features related to constitutional variables represents a potential constraint for using radiomic signals as cancer indicators. Our project exploits rigorous assessment of several radiomic features and we will correct those signals based on constitutional variables. (ii) The application of radiomic on ultrasonographic images: one of the main drawbacks of ultrasound is operator dependence, which may result in high levels of intraobserver and interobserver variability. However, a transvaginal ultrasound is the best method to identify and categorize endometrial tumors. One of the aims (aim#3) of our project is to assess the intraobserver and interobserver variability of radiomic features on ultrasonographic images, thus allowing us to reproduce our results in other clinical settings. Since TRACE4 has specific corrections allowing the harmonization of radiomic features from images measured with different imaging systems, we will assess the accuracy of such corrections in reducing or avoiding intraobserver and interobserver variability of radiomic features.

We expect that the radiomic analysis of ultrasonographic images by means of radiomic classifier of risks (TRACE4) will provide comparable results to molecular/genomic profiling. Hence, by adopting radiomic features instead of molecular/genomic profiling we will reduce money expenditure for the healthcare system. Additionally, image-based signatures will be an important tool for precise diagnosis and treatment, providing a powerful tool in modern medicine. In conclusion, the main goal of the project is to assess radiomics and radiogenomic features to identify endometrial cancer patients at high-risk. Adopting data characterization algorithms, we will replace the need to perform extensively molecular/genomic profiling in endometrial cancer [23,24,25,26]. Additionally, genomic/molecular data would be checked for the identification of new and emerging classes (eg, *HER2* or *PI3K/Akt/mTOR* pathway) of patients deserving of a personalized treatment [23,24,25,26].

## 4. Conclusions

The protocol of the present study aims to assess important questions in the management of endometrial cancer. In recent years, the adoption of molecular and genomic profiling has provided a practice change in the management of endometrial cancers. However, the costs and turnaround time associated with the execution of NGS represent the main barriers for the adoption of molecular testing. Moreover, we can speculate that in the future we will develop more complex prognostic scores involving a growing number of gene alterations. Evaluating radiomics and radiogenomic signatures would improve the process of disease characterization, reducing costs and turn-around time. Additionally, the adoption of radiomics might be useful in identifying new signals for a better personalization of the treatment.

## Data Availability

Not applicable.

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
