# Peer review of "Radiomics and Molecular Classification in Endometrial Cancer (The ROME Study): A Step Forward to a Simplified Precision Medicine"

_healthcare, 2022, doi:10.3390/healthcare10122464_

Round 1

Reviewer 1 Report

In their communication, Bogani  et al. briefly characterized the ROME study, in which the authors will do the attempt to assess radiomics and radiogenomics tools to identify high-risk EC patients /based on the Proactive Molecular risk classifier/. They hope that radiomics/radiogenomics may help to replace the extensive genetic profiling in patients with EC. The inclusion and exclusion criteria were well characterized and the study group of 100 EC consecutive patients will be selected at the Gynecology Unit of the Fondazione IRCCS Instituto Nazionale del Tumori di Milano. I hope the ROME study will be finalized in the forthcoming future and the eminent results will be available online.

Author Response

Comment: In their communication, Bogani  et al. briefly characterized the ROME study, in which the authors will do the attempt to assess radiomics and radiogenomics tools to identify high-risk EC patients /based on the Proactive Molecular risk classifier/. They hope that radiomics/radiogenomics may help to replace the extensive genetic profiling in patients with EC. The inclusion and exclusion criteria were well characterized and the study group of 100 EC consecutive patients will be selected at the Gynecology Unit of the Fondazione IRCCS Instituto Nazionale del Tumori di Milano. I hope the ROME study will be finalized in the forthcoming future and the eminent results will be available online.

Answer: We thank the reviewer for this comment. No changes are required. 

Reviewer 2 Report

This is a protocol paper. The authors plan to collect ultrasound images of 100 endometrial cancer cases and observe their correlation with molecular pathology classification. In addition, they will validate their predictive model using a validation cohort of 40 cases. They have three research specific objectives: #1 Evaluating the predictive value of radiomic feature in predicting specific mutational patterns in endometrial cancer patients; #2 Evaluated the scaled impact of radiomic signals; #3 Assess the intra-observer and inter-observer variability of radiomic signals obtained during clinical preoperative workup. While interesting, this reviewer would like to make a few comments in order to provide more detailed information to readers as a protocol paper.

#1. One of the research objectives of this study is to predict the molecular pathology classification of uterine cancer based on ultrasound imaging findings. Ultrasound images should be closely related to the gross and MRI findings of the tumor, and previous studies that have examined the relationship between this information and the molecular pathology classification of endometrial cancer should be described in the Introduction.

#2. Acquisition of ultrasound images requires a high degree of skill and is not highly reproducible between observers. Therefore, a more detailed description of what features/parameters of ultrasound images are used to construct a predictive model for molecular pathology classification is required. We would also like to see the rationale for the appropriateness of selecting these features/parameters. If it is tumor size, etc., MRI images would be able to provide more reproducible data.

Minor point:Line 130. What does [Ref nostra] mean?

Author Response

This is a protocol paper. The authors plan to collect ultrasound images of 100 endometrial cancer cases and observe their correlation with molecular pathology classification. In addition, they will validate their predictive model using a validation cohort of 40 cases. They have three research-specific objectives: #1 Evaluating the predictive value of radiomic feature in predicting specific mutational patterns in endometrial cancer patients; #2 Evaluated the scaled impact of radiomic signals; #3 Assess the intra-observer and inter-observer variability of radiomic signals obtained during clinical preoperative workup. While interesting, this reviewer would like to make a few comments in order to provide more detailed information to readers as a protocol paper.

Comment #1: One of the research objectives of this study is to predict the molecular pathology classification of uterine cancer based on ultrasound imaging findings. Ultrasound images should be closely related to the gross and MRI findings of the tumor, and previous studies that have examined the relationship between this information and the molecular pathology classification of endometrial cancer should be described in the Introduction.

Answer: In order to comply with the reviewer's comment, we discussed this in the introduction. 

Comment#2. Acquisition of ultrasound images requires a high degree of skill and is not highly reproducible between observers. Therefore, a more detailed description of what features/parameters of ultrasound images are used to construct a predictive model for molecular pathology classification is required. We would also like to see the rationale for the appropriateness of selecting these features/parameters. If it is tumor size, etc., MRI images would be able to provide more reproducible data.

Answer: One of the most interesting points of our protocol is that by adopting radiomics we aim to reduce variability. Evaluating a part of the tumor tissue, radiomics is able to identify the texture of the tumor thus providing information regarding the molecular status. In order to comply with the reviewer’s comment we clarified these points (specific aim 3 and experimental design 3)

Comment#3: Minor point:Line 130. What does [Ref nostra] mean?

Answer: We thank the reviewer for this comment. We corrected the reference

Reviewer 3 Report

The paper presents a fresh approach to acquiring  prognostic information leading to the development of new tools for the characterisation of EC and  tailored treatment.

Correlating radiomic signals with molecular/genomic profiling is novel. Using the  TRACE4© radiomic platform promises an inexpensive, fast method,  to be introduced as  soon as possible.

It is extremely encouraging that this area has been brought up for investigation. 

Author Response

Comment 1: The paper presents a fresh approach to acquiring  prognostic information leading to the development of new tools for the characterisation of EC and tailored treatment. Correlating radiomic signals with molecular/genomic profiling is novel. Using the  TRACE4© radiomic platform promises an inexpensive, fast method,  to be introduced as soon as possible. It is extremely encouraging that this area has been brought up for investigation. 

Answer: We thank the reviewer for this comment. No changes are required

Round 2

Reviewer 2 Report

The authors made adequate revisions according to the reviewer' comments and the issues have been clarified.